# Influence of hydrometeorological risk factors on child diarrhea and enteropathogens in rural Bangladesh

**Jessica A. Grembi** [1]*, **Anna T. Nguyen**[2], **Marie Riviere**[2], **Gabriella Barratt Heitmann**[2], **Arusha Patil**[2], **Tejas S. Athni**[3], **Stephanie Djajadi**[4], **Ayse Ercumen**[5], **Audrie Lin**[6], **Yoshika Crider**[7], **Andrew Mertens**[3], **Md Abdul Karim**[8], **Md Ohedul Islam**[8], **Rana Miah**[8], **Syeda L. Famida**[8], **Md Saheen Hossen**[8], **Palash Mutsuddi**[8], **Shahjahan Ali**[8], **Md Ziaur Rahman**[8], **Zahir Hussain**[8], **Abul K. Shoab**[8], **Rashidul Haque**[8], **Mahbubur Rahman**[8], **Leanne Unicomb**[8], **Stephen P. Luby**[1], **Benjamin F. Arnold**[9], **Adam Bennett**[10,11], **Jade Benjamin-Chung**[2,12]

**1** Division of Infectious Diseases and Geographic Medicine, Department of Medicine, School of Medicine, Stanford University, Stanford, California, United States of America, **2** Department of Epidemiology and Population Health, School of Medicine, Stanford University, Stanford, California, United States of America, **3** Harvard Medical School, Harvard University, Boston, Massachusetts, United States of America, **4** Division of Epidemiology, School of Public Health, University of California, Berkeley, Berkeley, California, United States of America, **5** Department of Forestry and Environmental Resources, North Carolina State University, Raleigh, North Carolina, United States of America, **6** Department of Microbiology and Environmental Toxicology, University of California, Santa Cruz, Santa Cruz, California, United States of America, **7** King Center on Global Development, Stanford University, Stanford, California, United States of America, **8** Infectious Disease Division, International Centre for Diarrhoeal Disease Research, Bangladesh, Dhaka, Bangladesh, **9** Francis I. Proctor Foundation and Department of Ophthalmology, University of California, San Francisco, San Francisco, California, United States of America, **10** Malaria Elimination Initiative, Global Health Group, University of California San Francisco, San Francisco, California, United States of America, **11** PATH, Seattle, Washington, United States of America, **12** Chan Zuckerberg Biohub, San Francisco, California, United States of America

* jgrembi@stanford.edu

**Data Availability Statement:** Not all data can be shared publicly due to Personally Identifiable Information (household GPS coordinates). However, de-identified data is publicly available on

## Abstract

### Background

A number of studies have detected relationships between weather and diarrhea. Few have investigated associations with specific enteric pathogens. Understanding pathogen-specific relationships with weather is crucial to inform public health in low-resource settings that are especially vulnerable to climate change.

### Objectives

Our objectives were to identify weather and environmental risk factors associated with diarrhea and enteropathogen prevalence in young children in rural Bangladesh, a population with high diarrheal disease burden and vulnerability to weather shifts under climate change.

### Methods

We matched temperature, precipitation, surface water, and humidity data to observational longitudinal data from a cluster-randomized trial that measured diarrhea and enteropathogen prevalence in children 6 months-5.5 years from 2012–2016. We fit generalized additive

the Open Science Framework (https://osf.io/yt67k/) and all code used for the analysis is available on GitHub at https://github.com/jgrembi/wbb-weather-diarrhea-enteropathogens. Analyses used R statistical software (version 4.2.0).

**Funding:** Research reported in this publication was supported in part by the Bill and Melinda Gates Foundation (https://www.gatesfoundation.org; to the University of California, Berkeley, under grant number OPPGD759 to SPL and to Stanford University, under grant number OPP1161946 to SPL). This work was supported by the National Institute of Allergy and Infectious Diseases of the National Institutes of Health (https://www.niaid.nih.gov) under Award Numbers K01AI141616 to JBC and R01AI166671 to BFA. JBC is a Chan Zuckerberg Biohub (https://www.czbiohub.org) Investigator. This work was also supported by National Heart, Lung, And Blood Institute of the National Institutes of Health (https://www.nhlbi.nih.gov) under award number T32HL151323 to ATN; by a Stanford University School of Medicine Dean's Postdoctoral Fellowship (https://med.stanford.edu) and by the National Institute of Diabetes and Digestive and Kidney Diseases of the National Institute of Health (https://www.niddk.nih.gov) under award number F32DK130574 to JAG; and by the National Institute of General Medical Sciences under grant number T32GM144273 to TSA. Funders had no role in the study design, data collection and analysis, decision to publish, or preparation of the manuscript.

**Competing interests:** The authors have declared that no competing interests exist.

mixed models with cubic regression splines and restricted maximum likelihood estimation for smoothing parameters.

## Results

Comparing weeks with 30˚C versus 15˚C average temperature, prevalence was 3.5% higher for diarrhea, 7.3% higher for Shiga toxin-producing *Escherichia coli* (STEC), 17.3% higher for enterotoxigenic *E. coli* (ETEC), and 8.0% higher for *Cryptosporidium*. Above-median weekly precipitation (median: 13mm; range: 0-396mm) was associated with 29% higher diarrhea (adjusted prevalence ratio 1.29, 95% CI 1.07, 1.55); higher *Cryptosporidium*, ETEC, STEC, *Shigella*, *Campylobacter*, *Aeromonas*, and adenovirus 40/41; and lower *Giardia*, sapovirus, and norovirus prevalence. Other associations were weak or null.

## Discussion

Higher temperatures and precipitation were associated with higher prevalence of diarrhea and multiple enteropathogens; higher precipitation was associated with lower prevalence of some enteric viruses. Our findings emphasize the heterogeneity of the relationships between hydrometeorological variables and specific enteropathogens, which can be masked when looking at composite measures like all-cause diarrhea. Our results suggest that preventive interventions targeted to reduce enteropathogens just before and during the rainy season may more effectively reduce child diarrhea and enteric pathogen carriage in rural Bangladesh and in settings with similar meteorological characteristics, infrastructure, and enteropathogen transmission.

### Author summary

Location-specific weather factors influence the fate, transport, and transmission of enteropathogens that cause diarrhea. We sought to identify hydrometeorological risk factors associated with childhood diarrhea and specific enteropathogens in rural Bangladesh. In this setting, higher temperatures and precipitation were associated with higher diarrhea in children under 5 years old and with several enteropathogens including bacteria, viruses, and parasites. Higher precipitation was associated with the lower prevalence of some enteric viruses. Our data suggest that interventions targeted to reduce pathogen transmission just before and during the rainy season might be most effective to reduce diarrhea in this setting.

## Introduction

Diarrhea is a leading cause of disability adjusted life years (DALYs) among children under 10 years old [1]. In low- and middle-income countries (LMICs), where the burden of diarrhea is high, asymptomatic enteropathogen carriage is also common [2] and is linked to child growth failure [2] and impaired child cognitive development [3]. While the global burden of diarrhea in diarrhea morbidity and mortality has steadily declined in LMICs [4], climate change poses a threat to further progress. Diarrhea risk is projected to increase by 15–20% due to climate change in the period 2040–2069 relative to the period 1961–1990 [5], and the World Health Organization estimates that additional deaths will be greatest among children from South Asia and Eastern Africa [6].

Because of the complex etiology of diarrhea, understanding relationships between weather and environmental risk factors and specific enteropathogen infection or asymptomatic carriage is critical to informing both current disease control efforts and climate change adaptation [5,7]. Enteric pathogens have differing environmental survival and transport, and relationships with weather (e.g. precipitation, temperature) and environmental factors (e.g. proximity of surface water to the household) are likely to be specific to particular pathogens [8]. Yet, there is limited evidence from LMICs linking these risk factors to a broad range of enteropathogens and diarrhea [9–12].

Prior studies have found increased diarrhea after heavy rainfall and floods, with at least some attribution given to the flow of fecal material into the environment from overwhelmed sewage treatment facilities in high-income contexts and latrine overflow/leakage in low- and middle-income contexts [13–15]. Parasitic and bacterial diarrhea are more common in the rainy season [12,16]. The concentration-dilution hypothesis posits that the greatest risk will occur at the beginning of a rainy season, after an antecedent period of no rainfall where pathogens can accumulate in the environment. Data have supported this hypothesis in most contexts [17]. Increased temperatures have been associated with higher incidence of bacterial diarrhea but lower incidence of viral diarrhea [16,18]. Higher temperatures are also associated with less *Escherichia coli* on children's hands, more cases of food-borne salmonellosis, and higher *E. coli* concentrations on food, in source water, and in stored drinking water in the household [19–21]. Higher relative humidity can increase the viability of bacteria but increase viral inactivation, while also increasing the transfer efficiency of both from fomites [22–24]. In LMICs, higher temperatures have been associated with a decrease in rotavirus prevalence and small increase in adenovirus prevalence; higher humidity and temperature have been associated with higher enteric bacteria prevalence and lower virus prevalence; and associations with precipitation varied by pathogen [9]. Furthermore, human behaviors such as handwashing and water treatment practices have been shown to vary under differing weather conditions. For example people are less likely to wash their hands but more likely to boil their drinking water during colder weather, and the use of water storage and treatment has been shown to increase in dry periods [25].

In Bangladesh, the prevalence of childhood diarrhea remains high, despite laudable progress in recent decades [26,27]. In addition, due to its low altitude, seasonal monsoon from June-September with resulting flooding, and high population density, Bangladesh is among the top ten countries most vulnerable to extreme weather events due to climate change [28]. In the absence of weather-informed enteropathogen mitigation strategies, Bangladesh's hard-earned reductions in diarrheal disease morbidity and mortality may be reversed. Yet, few studies have investigated relationships between weather, surface water, and enteric pathogen infection and carriage in a community setting in Bangladesh [29,30].

Here, we leverage community-level pathogen and diarrheal illness data from a large intervention trial in rural Bangladesh matched with fine-scale spatial resolution risk factors. Our objectives were to identify weather and environmental risk factors associated with diarrhea and enteropathogen prevalence in young children in order to inform targeted interventions for this vulnerable population. We hypothesize that there are pathogen-specific differences in the direction, magnitude, and significance of associations with hydrometeorological exposures.

## Methods

### Ethics statement

The trial was approved by ethical review committees at the International Centre for Diarrheal Disease Research, Bangladesh (icddr,b; PR- 11063), University of California, Berkeley (2011-

09-3652), and Stanford University (25863). Participants provided written, informed consent before enrollment and before fecal specimen collection.

## Study design

We analyzed data that was collected in a cluster-randomized trial of water, sanitation, hand-washing, and nutrition interventions in rural Gazipur, Mymensingh, Tangail, and Kishoreganj districts of Bangladesh (Fig 1) [31]. From 2012–2013, the trial enrolled a total of 720 village clusters spanning a geographic area of 12,500 km$^2$. The trial enrolled pregnant women and measured outcomes longitudinally in their children. On average, approximately 10% of the village was enrolled in each cluster [31] and the sample is generalizable to areas of mainland Bangladesh with similar ecological and socioeconomic features. This analysis was restricted to children enrolled in one of two cohorts of the WASH Benefits Bangladesh study: 1) the diarrhea cohort or 2) the enteropathogen cohort. These data span 3 years and were collected in one of six rounds (three rounds for the diarrhea cohort; and three rounds for the enteropathogen cohort which occurred ~6 months after each diarrhea cohort visit for children enrolled in this cohort). Each data collection round lasted roughly one year. Approximately 54 clusters were sampled per month from the diarrhea cohort, and 14 clusters were sampled per month from the enteropathogen cohort in a spatially dependent manner as the study team moved around the study area over the course of a year. Each round of household visits started in the same place and progressed along the same geographic path, thus compounds which were visited during the rainy season in the first year were more likely to be visited in the rainy season in subsequent years. Our pre-analysis plan is available at https://osf.io/f9cza/, and we note deviations from it in Appendix A in S1 Text.

## Outcomes

*Diarrhea*. Field workers measured caregiver-reported diarrhea in the previous 7 days (three or more loose or watery stools in a 24-hour period or a single stool with blood) in children 6

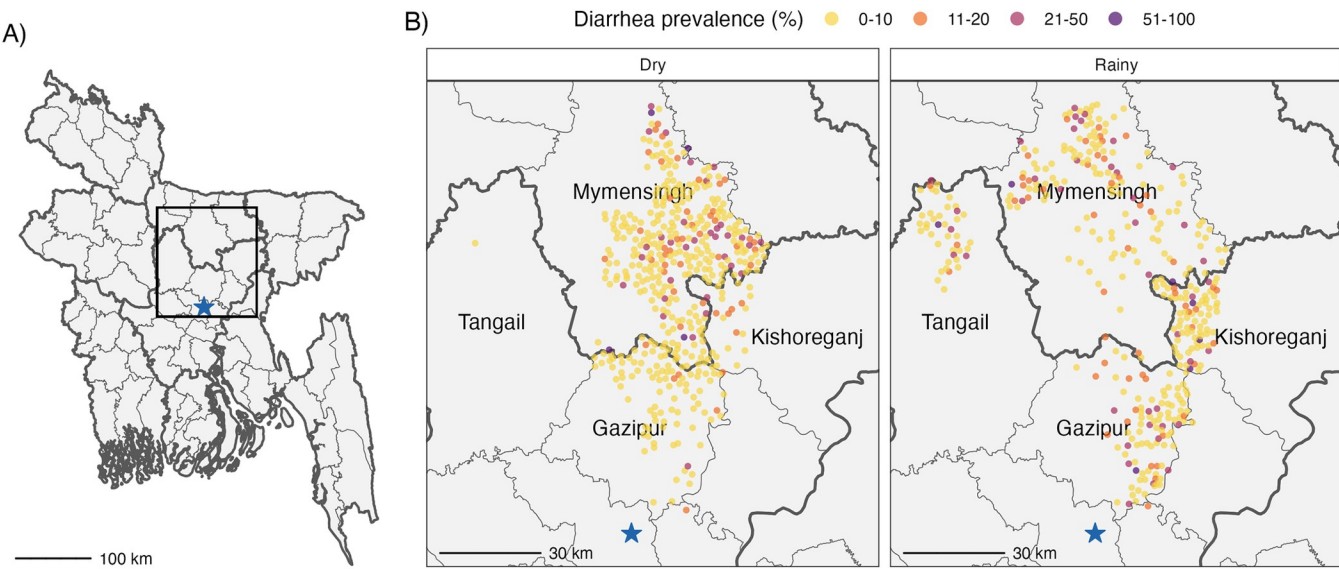

**Fig 1. Cluster-level diarrhea prevalence of study compounds in rural Bangladesh.** Panel A) situates the study area within Bangladesh. In panel B) points represent the centroid coordinates of a study cluster and are colored by cluster-level diarrhea prevalence within the past 7 days in the dry or rainy season. The blue star represents Dhaka, the capital of Bangladesh. Shapefiles were obtained from https://cran.r-project.org/web/packages/bangladesh/index.html.

months—5.5 years old (N = 4,478 children). The diarrhea cohort was restricted to the control arms of the original trial or data collected before intervention delivery (i.e. intervention arms at the baseline survey) (N = 7,320 measurements) because intervention effects on diarrhea are modified by weather [32]. Diarrhea prevalence was measured in three main rounds from the diarrhea cohort (N = 6,589 observations; May 2012—July 2013, September 2013—September 2014, December 2014—October 2015). This included index children (live births of women pregnant at the time of enrollment), other children living within the same compound that were younger than 3 years at study enrollment, and children within the same household born after the index children. Data on caregiver-reported diarrhea was collected in three additional rounds from children in the enteropathogen cohort (N = 731 observations from index children in the control arm; December 2012—January 2014, November 2013—November 2014, and March 2015—March 2016).

**Enteropathogens.**   The enteropathogen cohort included index children from a subsample of clusters that was evenly balanced across the control, WSH, nutrition, and N+WSH arms (allocation ratio 1:1:1:1). Clusters in each arm were selected based on logistical feasibility for specimen collection and transport to a central laboratory. Field workers collected stool samples from these children between November 2013 and November 2014 when they were approximately 14 months old and evaluated samples (N = 1,408) for 34 enteric viruses, bacteria and parasites using qPCR (details in Appendix B in S1 Text). Stool samples were collected as part of a routine household visit regardless of children's symptom status (14% of children reported diarrheal symptoms in the previous 7 days at the time of stool collection). We chose to collect stool from both asymptomatic and symptomatic children because prior literature has shown the importance of asymptomatic pathogen carriage on growth faltering [2,33,34].

We measured the following pre-specified primary outcomes: prevalence of 1) caregiver-reported diarrhea in the past 7 days, 2) any enteric virus (adenovirus 40/41, astrovirus, norovirus GI/GII, rotavirus, sapovirus), and 3) any parasite (*Cryptosporidium* spp, *Enterocytozoon bieneusi*, or *Giardia* spp). We did not include prevalence of any bacteria because at least one bacterium was detected in over 95% of samples. Pathogen-specific prevalence for pathogens detected in >10% of samples were secondary outcomes.

To detect potential unmeasured confounding or outcome measurement bias in the diarrhea analysis, we repeated the analysis using caregiver-reported child bruising in the past 7 days as a negative control outcome [35,36].

## Hydrometeorological variables

We matched hydrometeorological risk factors from remote sensing datasets to trial data by date of outcome measurements and geocoordinates of study compounds (groups of households in which patrilineal families share a common courtyard). Two of the exposures (temperature and precipitation) had daily resolution data and the other (surface water and humidity) had monthly resolution. Those with daily resolution were aggregated in different ways over a 7-day window (as well as a 30-day window for temperature) and lagged by 0, 1, 2, or 3 weeks (corresponding to the periods 1–7, 8–14, 15–21, and 21–28 days before outcome measurement). As diarrhea was measured at a weekly resolution (any diarrhea in the previous 7 days), we included 1- to 3-week lags for diarrhea and 0- to 3-week lags for pathogen outcomes per previous studies reporting these as relevant lag times between temperature or precipitation and either diarrhea or enteric pathogens [9,18,37,38]. The lag periods include expected time for microbial growth/death due to weather (e.g. temperature and humidity), mobilization and transport of enteropathogens persisting in the environment (e.g. after precipitation), and post-exposure enteropathogen-specific incubation times within the host to reach detectable levels

and/or initiate a diarrheal response (Appendix C in S1 Text). Exposures with monthly resolution were linked to the outcomes by calendar month of outcome ascertainment without any lag period. Here we briefly summarize each variable; Appendix D in S1 Text includes additional details.

**Temperature.** Using 0.001 degree (~1 km) resolution daily near surface air temperature data from the National Aeronautics and Space Administration Famine Early Warning System Network Land Data Assimilation System (FLDAS) Central Asia dataset [39], we calculated the average, minimum and maximum temperature values for 7 and 30 days prior to outcome observation, with 1-3-week lags for diarrhea and 0-3-week lags for pathogen outcomes.

**Precipitation.** We obtained 0.1 degree (~10 km) resolution daily precipitation data from the Multi-Source Weighted-Ensemble Precipitation dataset from GloH2O which merges gauge, satellite, and reanalysis data and corrects for bias [40]. We created an indicator for heavy rainfall (any day in the prior week with total daily precipitation >80th percentile [17mm] on rainy days during the study period) and an indicator for whether the weekly sum of precipitation (over the past 7 days) was above or below the median (13mm) for the entire study period. Precipitation thresholds were based on the percentile of all daily totals (across all study years and not season-specific) in order to maximize the generalizability of our results. Each variable was calculated with 1-3-week lags for diarrhea and 0-3-week lags for pathogen outcomes. As a sensitivity analysis for the thresholds to define the categorical variables, we also include a heavy rainfall variable calculated using the 90th percentile (29mm), and weekly sum of precipitation above the 75th (58mm) and 90th (105mm) percentiles. Although prior studies have used different fixed periods between the months of June and October as the rainy season in Bangladesh [30,41,42], we observed variations in the start and end of the rainy period in different years of our study, with substantial rainfall before June in every year. Therefore, we determined the rainy season empirically [10], defined as the continuous period during which the 5-day rolling average of daily precipitation was ≥10mm/day.

**Surface water.** We obtained monthly 30 m resolution surface water data for the period 1984–2020 from the Global Surface Water Explorer [43]. Variables included: 1) seasonal surface water consistently present in a season, 2) ephemeral surface water present intermittently, and 3) any surface water (ephemeral, seasonal, or permanent) detected in the month of outcome measurement, including man-made and natural water bodies. We calculated tertiles of distance from each household to the nearest surface water and created an indicator for whether the proportion of pixels with surface water within 250m, 500m, or 750m of each household was above or below the median. The distance thresholds were determined to capture risks associated with the potential for flooding in the household environment during heavy rainfall.

**Humidity.** We obtained mean monthly vapor pressure deficit data with 4 km resolution from Terraclimate [44] and measured associations with a continuous measure in kilopascals (kPA).

## Statistical analysis

We used generalized additive mixed models with a Poisson family with a log link to estimate the relationship between exposures (continuous exposures modeled with cubic splines, categorical exposures modeled as factors relative to a reference category) and outcomes [45–47]. To estimate simultaneous confidence intervals, we resampled from the variance-covariance matrix under a multivariate normal distribution [48]. To assess potential spatial autocorrelation, we used Moran's I coefficient and included a bidimensional thin plate spline function of household latitude and longitude for models in which we detected spatial autocorrelation. Due to multiple measurements within a study compound in the diarrhea dataset (e.g. index

children and siblings sampled within a single compound on the same day and data collected over multiple years), we included random intercepts for study compound for diarrhea models. In the pathogen dataset, only a single measurement was collected for each index child thus we included random intercepts for study cluster for pathogen outcomes to account for village-level clustering. For a small number of models evaluating categorical exposures, there was insufficient spatial variation leading to poor model fit for the generalized additive mixed models. In this case, we reran the models as generalized linear models with a Poisson family and log link without spatial terms and used robust standard errors to account for clustering. We adjusted for covariates that were associated with each outcome in our dataset using a likelihood ratio test (p-value < 0.1) (details in Appendix E in S1 Text). For pathogen carriage outcomes, we pooled across all study arms in which specimens were collected (N = 1,408 measurements) to maximize statistical power and included receipt of WASH or nutrition interventions as covariates; this was not necessary for the diarrhea outcome as only observations from the control arms were used. In cases when the number of covariates selected through screening was large and would have resulted in poor model fit due to data sparsity, we removed covariates from the adjustment set sequentially, starting with those with the weakest association to the outcome. We predicted prevalence from adjusted models with continuous covariates set at the median across all samples, intervention set at receiving both WASH and nutrition interventions, household wealth set at the lowest quartile, antibiotic use set at none in the previous 7 days, and sex set as male.

We assessed effect modification for the diarrhea outcome by child age (<1.5 vs. ≥1.5 years) because age is strongly associated with diarrhea risk [49,50]. Because pair-wise correlations between temperature and precipitation were strong (S1 Fig), we evaluated temperature exposure models with and without inclusion of precipitation as an additional covariate. We also assessed effect modification of temperature by precipitation using a model with both exposures and an interaction term. Neither vapor pressure deficit nor surface water displayed correlations with other hydrometeorological variables, so we did not adjust these exposure models for other exposures.

## Results

### Study participant characteristics

The diarrhea cohort included 4,478 children (mean age = 23 months; SD = 12m) measured in three rounds from 2012–2016, and the pathogen cohort included a subset of 1,408 children (mean age = 14 months; SD = 2) measured primarily in 2014 (Table 1). Other demographic characteristics of each cohort have previously been reported [31,51]. Overall, diarrhea prevalence was 7.1%, with 6.1% prevalence in the dry season and 8.3% in the rainy season (Fig 1B; prevalence ratio of rainy vs dry season 1.36, 95% CI 1.14, 1.62).

### Temporal trends

The study period spanned three rainy seasons from 2013–2015. During the rainy season, average weekly temperature was higher (Fig 2). Distance to surface water was slightly lower during the peak of each rainy season. Diarrhea measurements were not collected in some weeks of the 2013–14 rainy seasons.

### Temperature

During the study period, the average weekly temperature across the study area ranged from 15–33˚C, the minimum weekly temperature ranged from 13–33˚C, and the maximum weekly

**Table 1. Study participant and sample characteristics.**

|  | Diarrhea cohort | Pathogens cohort |
|---|---|---|
| Children | 4,478 | 1,408 |
| Observations | 7,320 | 1,408 |
| **Year of measurement** | | |
| 2012 | 491 (6.7%) | 0 (0.0%) |
| 2013 | 2,680 (36.6%) | 19 (1.3%) |
| 2014 | 1,868 (25.5%) | 1,389 (98.7%) |
| 2015 | 2,170 (29.6%) | 0 (0.0%) |
| 2016 | 111 (1.5%) | 0 (0.0%) |
| Mean age, months (SD) | 22.6 (11.8) | 14.0 (2.0) |
| Antibiotics consumed in the past week | –[a] | 233 (16.5%) |
| **Season** | | |
| Rainy[b] | 3,408 (46.6%) | 620 (44.0%) |
| Dry | 3,912 (53.4%) | 788 (56.0%) |
| **Intervention arm in original trial** | | |
| Control[c] | 7,320 (100.0%) | 328 (23.3%) |
| WASH[d] | 0 (0.0%) | 368 (26.1%) |
| Nutrition | 0 (0.0%) | 352 (25.0%) |
| Nutrition + WASH[d] | 0 (0.0%) | 360 (25.6%) |

[a] Not measured

[b] Defined as the period when the 5-day rolling average rainfall was ≥10mm/day.

[c] This includes 2066 baseline measurements (before intervention delivery) from children whose households were randomized to later receive an intervention.

[d] Water, sanitation, and handwashing

temperature ranged from 17–34˚C. Enteropathogens were measured during a one-year period of the study, during which the minimum temperatures did not fall below 17˚C and the average weekly temperature did not fall below 19˚C.

According to adjusted model results using a 1-week lag, diarrhea prevalence was 3.3% (95% CI 2.0%, 5.6%) at average weekly temperature of 15˚C and 6.8% (95% CI 5.2%, 8.9%) at 30˚C; unadjusted models produced similar results (prevalence of 2.9% [95% CI 1.8%, 4.7%] at 15˚C and 5.8% [95% CI 4.9%, 7.0%] at 30˚C). Trends remained evident at longer lags, and the rate of increase in diarrhea prevalence was higher at weekly average temperatures above 30˚C (Fig 3A). Higher average weekly temperatures were also associated with higher prevalence of Shiga toxin-producing *E. coli* (STEC) (S1 Table), enterotoxigenic *E. coli* (ETEC) (S1 Table), and higher *Cryptosporidium* (8.0% [95% CI 3.8%, 16.7%] at 19˚C; 16.0% [95% CI 11.5%, 22.1%] at 30˚C; 2-week lag using adjusted models) (Fig 3C and 3E). Sapovirus prevalence displayed an inverse relationship, with 15.9% prevalence (95% CI 6.4%, 39.3%) at average weekly temperatures of 19˚C and 7.1% (95% CI 4.6%, 11.2%) at 30˚C (1-week lag using adjusted models). These models all appeared linear, thus we can infer that each 1˚C increase in temperature was associated with approximate prevalence increases of 0.23% for diarrhea, 0.66% for STEC, 1.57% for ETEC, and 0.73% for *Cryptosporidium*, and a 0.80% decrease for sapovirus. The relationship between temperature and the prevalence of *Enterocytozoon bieneusi*, *Giardia*, and norovirus was nonlinear (Fig 3D and 3E), but the confidence intervals were wide. There was no association between temperature and the prevalence of other bacteria (Fig 3B) or other viruses (Fig 3D). Associations between temperature and the combined metrics of 'any virus' or 'any parasite' masked the heterogeneous effects of individual viral or parasitic enteropathogens

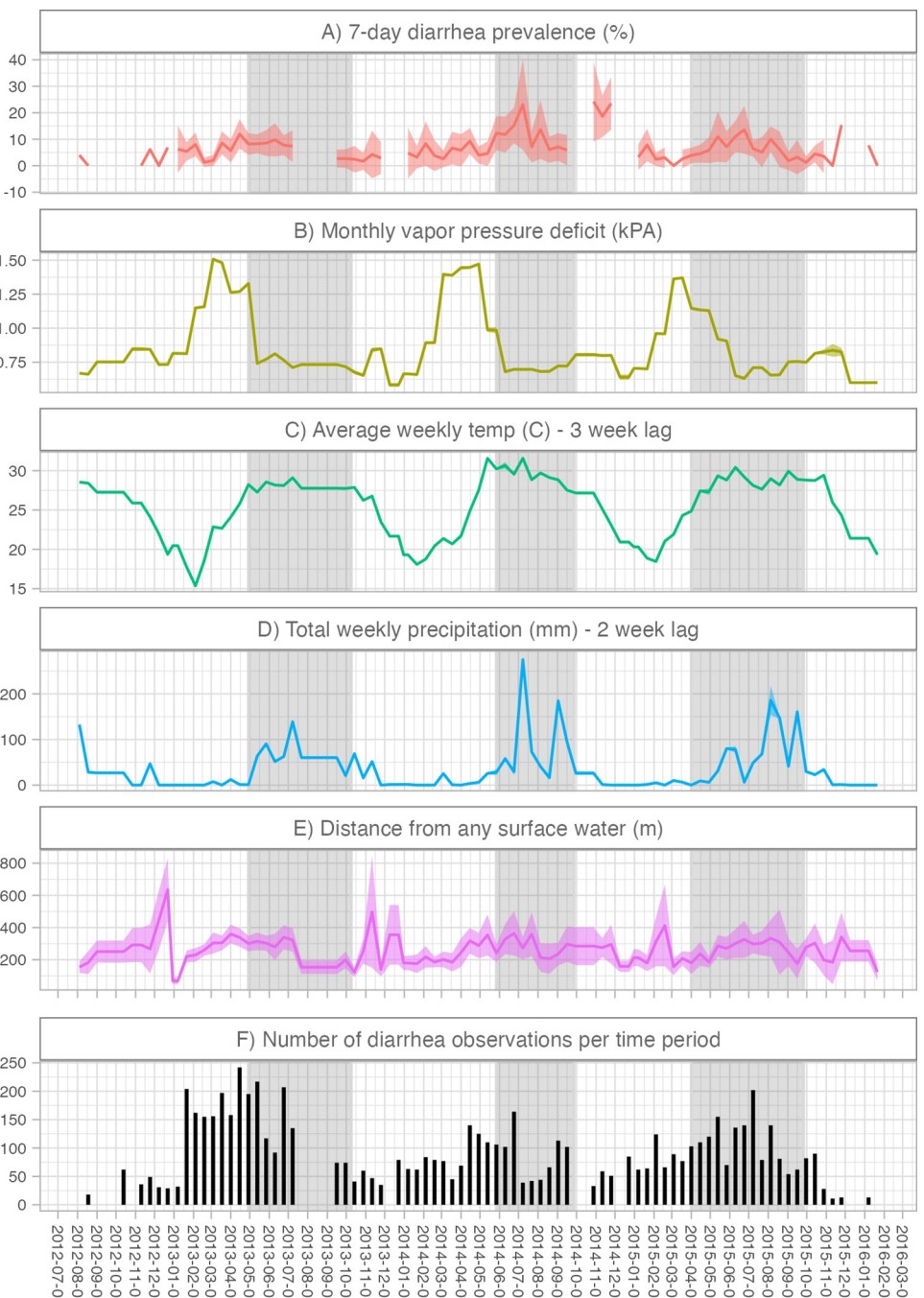

**Fig 2. Time trends in risk factors, diarrhea prevalence, and number of observations.** Shaded bands indicate 95% confidence intervals using robust sandwich standard errors to account for correlation within study clusters. Estimates exclude biweekly periods when the number of observations was <10. Panel A) includes diarrhea measurements from the control arms of the original trial. Breaks in the line indicate periods when the study did not collect data on diarrhea status. In panels B) to E), we display the risk factor values that corresponded to the diarrhea measurements in our analysis. For periods when there was no diarrhea data collection, we display the risk factor values for the next biweekly period with at least 10 diarrhea measurements. Panel F) displays the number of diarrhea observations in each biweekly period. Rainy seasons (grey shaded regions) were determined empirically as periods where the 5-day rolling average rainfall was ≥10mm/day.

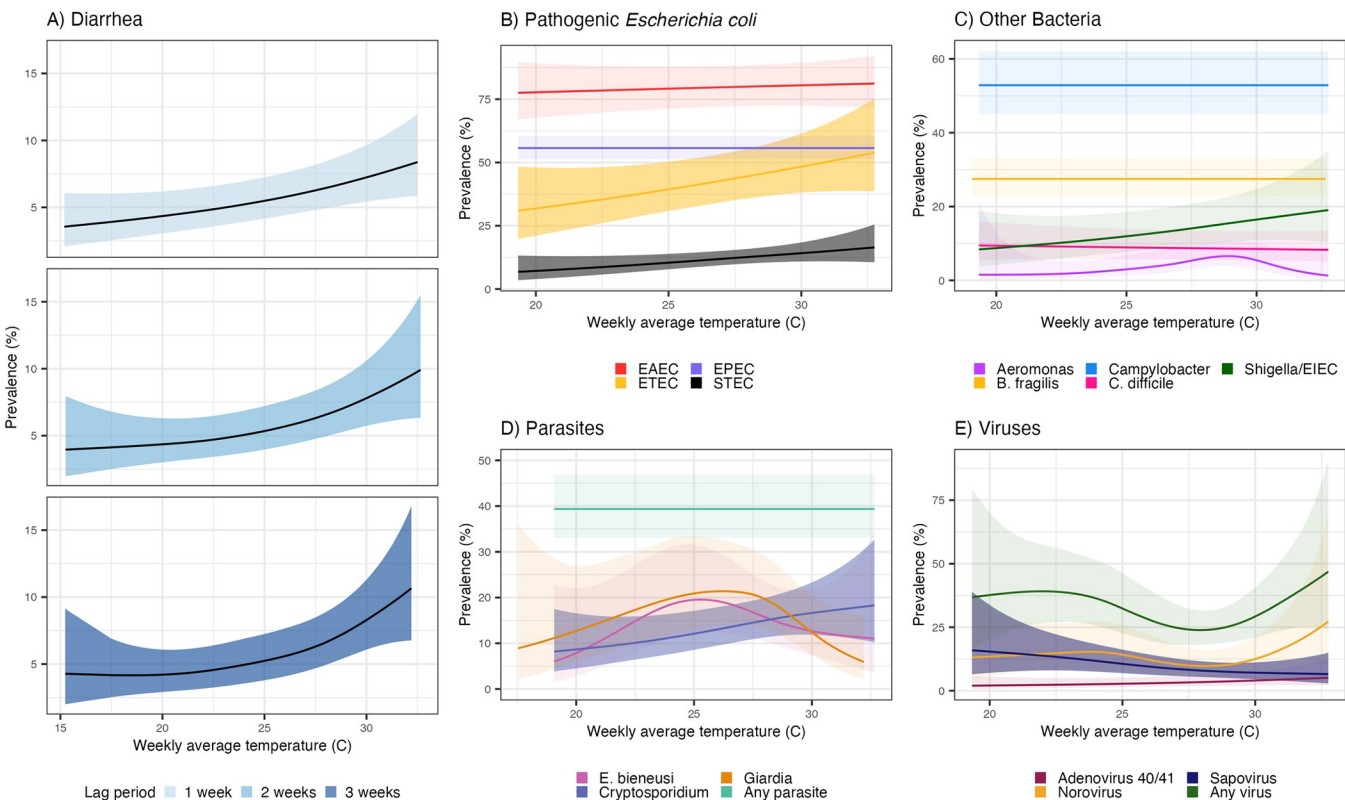

**Fig 3. Diarrhea and enteropathogen prevalence by weekly average temperature.** All panels present adjusted models (see Appendix E in S1 Text for details); shaded bands indicate simultaneous 95% confidence intervals (CIs) accounting for clustering. Panel A) includes diarrhea measurements in children aged 6 months—5.5 years in the control arms in the original trial. Panels B-E) include measurements in children approximately 14 months of age in the control, combined water + sanitation + handwashing (WASH), nutrition, and combined nutrition + WASH arms of the original trial. We present results for the lag periods that best aligned with the incubation period for each outcome (Appendix C in S1 Text); results with alternative lag periods are in S2 Fig. S1 Table provides point estimates and 95% CIs for bacteria (panels C-D) at 19°C and 30°C. Prevalence estimates were predicted under conditions which held all adjustment covariates at fixed representative values (see Methods for details).

(Fig 3D and 3E). Overall, results for diarrhea and pathogen prevalence were similar using alternative lag periods and with stronger effects at 2- and 3-week lags for bacterial pathogens (Figs 3A and S2) and using minimum or maximum weekly temperature (S3–S5 Figs).

## Precipitation

Median total weekly precipitation during the study period was 13mm (range: 0-396mm), with medians of 0mm (range: 0-198mm) in the dry season and 63mm (range: 0-396mm) in the rainy season. Above-median weekly total precipitation (>13mm) was associated with higher diarrhea prevalence (PR = 1.21; 95% CI 0.98, 1.51; adjusted PR [aPR] = 1.29, 95% CI 1.07, 1.55) using a 2-week lag, but not using 1- or 3-week lags (Fig 4A) nor when using higher threshold cutoffs (75th and 90th percentile, S6 Fig). For parasites, above-median weekly precipitation was associated with higher *Cryptosporidium* prevalence (e.g., aPR = 2.07; 95% CI 1.39, 3.10 for a 3-week lag), but lower *Giardia* prevalence (e.g., aPR = 0.66; 95% CI 0.49, 0.90 for a 2-week lag) (Fig 4B). For viruses, above-median weekly precipitation was associated with higher prevalence of adenovirus 40/41, but lower prevalence of sapovirus and norovirus (Fig 4C). For bacteria, above-median weekly precipitation was associated with higher prevalence of enterotoxigenic *E. coli* (ETEC), STEC, *Shigella*/EIEC, *Campylobacter*, and *Aeromonas* (Fig 4D). With higher percentile cutoffs, results were similar for viruses, and similar but attenuated

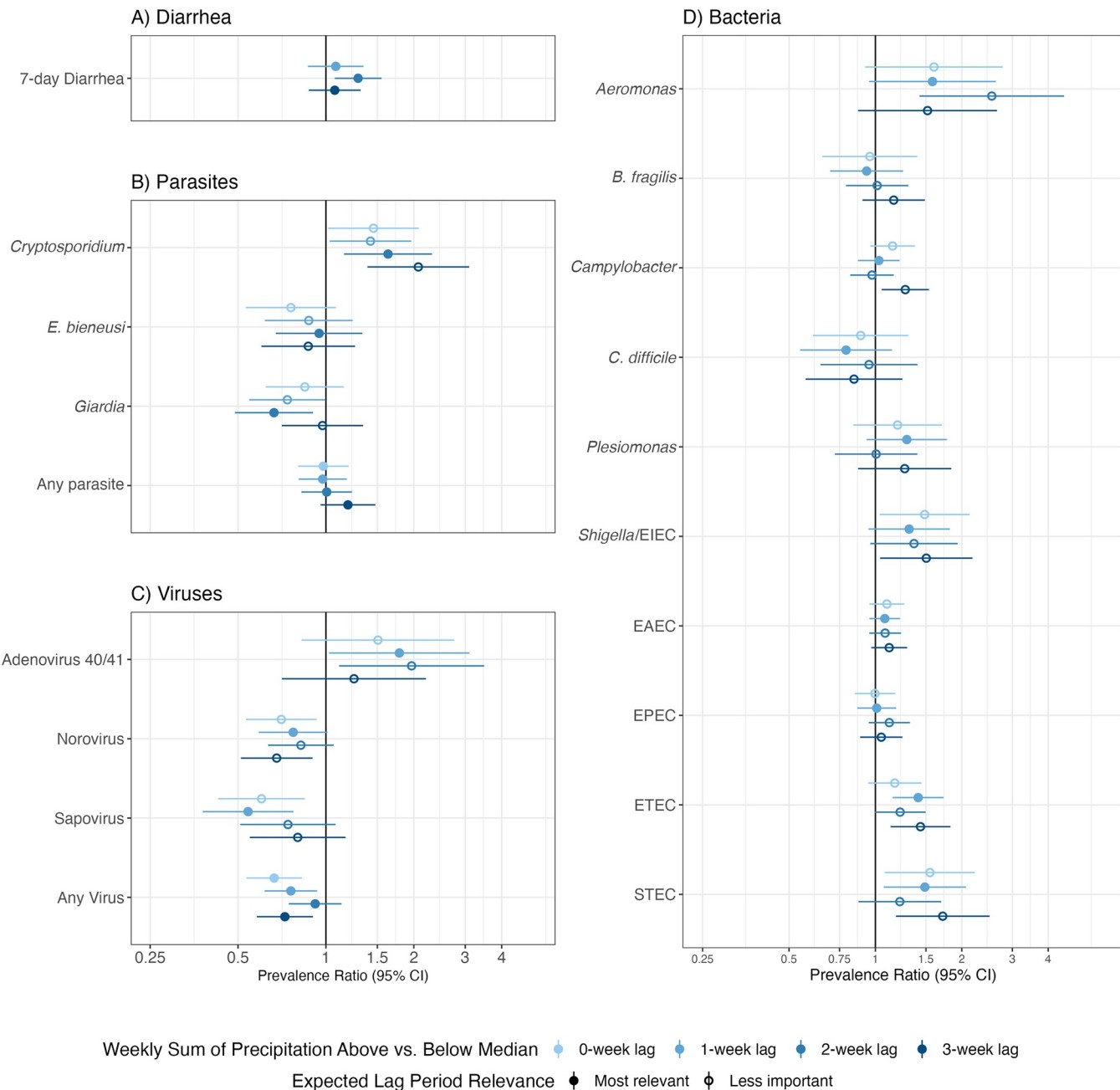

**Fig 4. Association between diarrhea and enteropathogen prevalence and above- vs. below-median average weekly precipitation.** All panels present adjusted models including an indicator for above median (13mm) average weekly precipitation as the independent variable; unadjusted models produced similar results. Error bars present 95% confidence intervals adjusted for clustering. The x-axis is on the log scale. Panel A) includes diarrhea measurements in children aged 6 months—5.5 years in the control arms in the original trial. Panels B-D) include measurements in children approximately 14 months of age in the control, combined water + sanitation + handwashing (WASH), nutrition, and combined nutrition + WASH arms of the original trial. Closed circles in panels B-D indicate the expected most important lag based on enteropathogen-specific incubation times (Appendix C in S1 Text).

for bacteria and parasites except for *E. bieneusi*, for which increased precipitation was associated with a lower prevalence using both the 75th and 90th percentile cutoffs with a 1-week lag (S6 Fig). Heavy rainfall (>80th percentile, 17mm) on a single day was associated with higher prevalence of diarrhea (aPR = 1.31, 95% CI 1.09, 1.57; 2-week lag), *Cryptosporidium*,

adenovirus 40/41, *Aeromonas*, *Campylobacter*, *Shigella*/EIEC, ETEC and lower prevalence of norovirus and sapovirus (S7 Fig). Results were similar with a 90[th] percentile cutoff threshold for all outcomes except diarrhea where the association was lost (>29mm; S7 Fig). Results varied by lag period, and the lag period expected to be most important based on enteropathogen-specific incubation times (Appendix C in S1 Text) were often not the most strongly associated with above-median precipitation or heavy rainfall (Figs 4B–4D and S6 and S7).

## Surface water

The mean distance from study households to the closest surface water was 277m (range: 11-1818m) for any surface water, 446m (range: 11–1852) for ephemeral surface water, and 396 (range: 11–1839) for seasonal surface water. Compared to households in the highest tertile of distance to any surface water (> 315m), *Aeromonas* prevalence was lower for those in the middle tertile (165-315m) (aPR = 0.53; 95% CI 0.29, 0.96) and in the lowest tertile (<165m) (aPR = 0.43; 95% CI 0.23, 0.81) (S8 Fig). *Aeromonas* results were similar for ephemeral and seasonal surface water. Distance to surface water was not associated with other outcomes.

Around each study household, the median proportion of land that contained any surface water was 0.4% (range: 0–76%) within 250m, 4% (range: 0–76%) within 500m, and 6% (range: 0–76%) within 750m. Most outcomes were not associated with the proportion of land that contained surface water. For certain radii and surface water types, an above-median proportion of surface water was associated with lower diarrhea, *Aeromonas*, adenovirus 40/41, norovirus, and *E. bieneusi* prevalence and higher prevalence of *Shigella*/EIEC; however, most confidence intervals were close to or spanned the null (S9 Fig).

## Humidity

Vapor pressure deficit (VPD) ranged from 0.49 to 1.71 kPa (mean 0.86 kPa) during the study period. Higher values of VPD, corresponding to lower humidity, were associated with decreases in ETEC (S10 Fig); prevalence was 41.6% (95% CI 26.8%, 64.5%) at VPD of 0.66 kPa and 16.1% (95% CI 8.1%, 32.0%) at VPD of 1.56 kPa according to the adjusted model. Higher VPD was also associated with lower prevalence of Shigella/*EIEC*, sapovirus, and any virus (S10 Fig). No associations with other outcomes were observed.

## Other analyses

We detected meaningful effect modification in our pre-specified subgroup analyses evaluating child age category (6 months-1.5 years, 1.5–5 years) for the diarrhea outcome. Diarrhea in children aged 1.5–5 years old tended to be impacted by precipitation to a greater extent than younger children (S11 Fig). Heavy rain significantly increased diarrhea prevalence in older children (aPR 1.54, 95% CI 1.21, 1.97) but not those in the younger group (aPR 1.07, 95% CI 0.83, 1.37) in adjusted models with a 3-week lag (results were similar with a 2-week lag).

Due to the strong correlation between temperature and precipitation, we included models with both temperature and precipitation variables. Models which included both variables often had improved model fit (lower Akaike Information Criterion) over models with only temperature. There were few instances in which models were improved by the inclusion of an interaction term (S2 Table).

Our negative control analysis using caregiver-reported child bruising found null associations with most risk factors and a small association in the opposite direction than we observed for diarrhea for a few of the temperature risk factors, suggesting that potential misclassification of the outcome did not substantially influence our results (S3 Table and S12 Fig).

## Discussion

In this analysis of the association between weather and environmental variables and enteric pathogen carriage in children in rural Bangladesh, we found that higher weekly average temperatures and above-median precipitation were associated with higher prevalence of diarrhea and certain enteric pathogens. Surface water presence near the household was also associated with carriage of certain enteric bacteria and parasites. Consistent with prior studies, our analysis found that the effects between hydrometeorological variables and enteropathogens are heterogeneous [9,14,16]. We emphasize the heterogeneity even within pathogen class, such that effects can be masked when looking at composite variables (e.g. any viral enteropathogen) or even at the level of all-cause diarrhea.

### Precipitation and diarrhea

Above-median weekly precipitation with a 2-week lag was associated with up to 29% higher diarrhea prevalence and heavy rainfall with a 2-week lag was associated with 31% higher prevalence of diarrhea. A prior meta-analysis found both positive and negative associations between precipitation and diarrhea risk [17]. Generally, heavy rainfall preceded by a dry period has been associated with higher diarrhea risk, while heavy rainfall preceded by a wet period has been associated with lower diarrhea risk. This may be because pathogens concentrate during dry periods and then are flushed into the environment when heavy rainfall occurs (the concentration-dilution hypothesis) [17]. We were not able to investigate associations with heavy rainfall preceded by a dry vs. wet period because almost all heavy rainfall periods were preceded by rainy days. Taken together, our findings suggest that in our study setting, even moderately higher precipitation was associated with meaningful increases in diarrhea. However, these results obscure important heterogeneous impacts of precipitation on specific enteropathogens, which we discuss below.

### Precipitation and enteropathogens

We found that above-median (>13mm) total weekly precipitation was associated with a higher prevalence of *Cryptosporidium*, adenovirus 40/41, *Aeromonas*, *Shigella*/EIEC, *Campylobacter*, ETEC, and STEC and lower prevalence of norovirus, sapovirus, and *Giardia*. Thus, five of the top ten diarrhea-attributable pathogens for children under 24 months old displayed positive associations with above-median total weekly precipitation [49]. A recent meta-analysis of data from children up to 6 years old in 19 LMICs with tropical climates found that higher precipitation was associated with a small decrease in ETEC and *Campylobacter* prevalence and no difference in *Cryptosporidium*, *Shigella*, *Giardia*, or enteric virus prevalence [9]. Pathogen prevalence was similar for viruses, slightly higher for *Campylobacter* and *Cryptosporidium*, and lower for Giardia and *Shigella/EIEC* in our study compared to the prior meta-analysis. Some of these differences are likely due to the narrower age range of children included in our pathogen cohort (approximately 14 months old) compared to those in the meta-analysis, particularly for *Giardia* which is typically seen in older children. Additionally, our pathogen cohort captured primarily asymptomatic enteropathogen carriage within the community (only 14% of children had reported diarrhea) while the previous meta-analysis relied heavily on hospital-based surveillance and case-control studies of overt diarrheal illness. Our findings may also differ due to varying urbanicity and WASH infrastructure [52] compared to the locations included in the meta-analysis.

Interestingly, some enteropathogen associations with precipitation were stronger for lag periods that differed from the lag period we expected to be most relevant based on within-host incubation times. Particularly for bacteria, analyses using 2- and 3-week lag periods had

stronger associations than those with a 1-week lag; this may reflect higher environmental growth, survival, and/or delayed transport (e.g. from flooding that occurs days after a rain event) following higher precipitation. Adenovirus 40/41, a double-stranded DNA (dsDNA) virus, displayed an inverse relationship with precipitation compared to the other viruses evaluated, which were both single-stranded RNA (ssRNA) viruses. A prior study showed that ultraviolet irradiation led to a 4-$\log_{10}$ inactivation of ssRNA viruses but only a 2-$\log_{10}$ inactivation of dsDNA viruses in water [53]. It is possible that increased water turbidity associated with precipitation events intensifies this relationship. Estimating environmental survival of enteric pathogens under different weather conditions is an important area for future research.

## Temperature

For temperature, our estimated associations were more modest overall than those from previous studies. One meta-analysis–encompassing populations of all ages from various climate regions–estimated that a 1˚ C increase in mean temperature was associated with relative increases of 7% for diarrhea and bacterial diarrhea and no association with viral diarrhea [16]. Another study focusing on children from LMICs estimated that increasing weekly average temperatures in the range between 10–40˚ C was associated with higher risk of *Campylobacter*, ETEC, *Shigella*, *Cryptosporidium*, *Giardia*, and adenovirus and lower risk of sapovirus and rotavirus, and generally associations were stronger [9]. Our more modest estimates may reflect the fine-scale spatial resolution of our study or differing background levels of enteropathogen transmission. Our findings reinforce that relationships between temperature and enteropathogen carriage are heterogeneous, even within pathogen class (i.e. virus, bacteria, parasite). The null results for the composite measures of 'any virus' or 'any parasite' in the background of significant effects for individual viruses/parasites cautions against evaluating higher-level groupings that may mask pathogen-specific effects.

## Limitations

This study was subject to several limitations. First, in 2014, the original trial did not collect data during peak rainy season (Fig 2), limiting our ability to make inferences about associations with heavy rainfall in that season. Second, we only measured enteropathogens in a subsample, so statistical power was limited for certain analyses; for this reason, we did not consider it feasible to estimate associations with class-specific diarrhea (e.g., bacterial diarrhea) as prior studies have done. However, measuring enteropathogen carriage in a community-based cohort (including asymptomatic and symptomatic infections) sheds light on weather-specific transmission patterns that may not be discernable when restricting to symptomatic cases and individuals seeking care at health facilities. Third, the enteropathogen sample included a narrow age range (approximately 14 months); results may not generalize to other ages. Rotavirus, a highly climate-sensitive pathogen, was not included in our pathogen-specific analysis because it did not meet our 10% prevalence threshold (although detection of rotavirus is accounted for in our composite "Any virus" variable). Rotavirus was detected in only 2.2% of our community-based cohort, which is similar to other cross-sectional studies where the majority of samples are not associated with diarrheal symptoms. Our effect modification analyses may also have had limited statistical power. Finally, because relationships between hydrometeorological factors, diarrhea, and enteropathogen infection vary by climate zone [54], our findings might not generalize to settings with differing enteropathogen transmission levels, WASH infrastructure, or hydrometeorological characteristics.

## Climate and policy implications

Climate change models predict increases in temperature (mean increase of 1˚ C) and precipitation (mean increase of 6%) in Bangladesh from 2021–2050 [55]. Our findings suggest that under these projections, there would be small increases in the prevalence of diarrhea and certain enteric pathogen infections.

One potential implication of our findings is that interventions to reduce enteropathogen infections in Bangladesh and similar settings may be most beneficial prior to or during the rainy season. A recent analysis found that WASH interventions were more effective at reducing diarrhea during periods of heavy precipitation [32,56]. Seasonal targeting of interventions is a common and cost-effective strategy for other diseases with seasonal transmission patterns, such as malaria [57], but has yet to be used for enteric illness. Of note, the enteropathogens that had higher prevalence under higher temperature and precipitation levels are not currently vaccine-preventable. Many can be prevented by household WASH interventions [51], but sustained use is critical to realizing health benefits [58,59] and has proven difficult to maintain in the long term [60–62]. Given the high cost and user burden of household WASH interventions [60], seasonal targeting via the direct provision of resources and/or increased promotion may improve both public health impact and cost-effectiveness.

## Conclusions

We observed heterogeneous impacts of weather on community-level enteropathogen carriage by pathogen class and species for young children in rural Bangladesh. However, the prevalence of a majority of the enteropathogens, as well as diarrheal illness, displayed a positive association with total weekly precipitation above 13mm. In similar settings, preventive interventions targeted at the beginning of the rainy season may be an effective strategy for reducing enteric pathogen infections and carriage, particularly under climate change.

## Supporting information

**S1 Text. Appendices.** Includes: A. Deviations from pre-analysis plan. Appendix B. Detection of enteropathogens in stool. Appendix C. Incubation periods for each pathogen outcome. Appendix D. Description of risk factors and data sources. Appendix E. Covariate sets tested for each risk factor.
(PDF)

**S1 Fig. Pairwise relationships between weather variables.** Bivariate scatter plot of continuous climatic risk factors in the diarrhea cohort. Correlation ellipses depict the strength of the association on the basis of the Spearman rank correlation, color of the ellipse indicates the direction of the correlation, and the correlation coefficient is printed inside each ellipse.
(PDF)

**S2 Fig. Predicted enteropathogen prevalence by weekly average temperature (C) with different lags.** All panels present adjusted models for children approximately 14 months of age in the control, combined water + sanitation + handwashing (WASH), nutrition, and combined nutrition + WASH arms of the original trial. Shaded bands indicate simultaneous 95% confidence intervals accounting for clustering. Prevalence estimates were predicted under conditions which held all adjustment covariates at fixed representative values (see Methods for details).
(PDF)

**S3 Fig. Predicted diarrhea prevalence by temperature minimum, mean and maximum.** All panels present adjusted models for temperature with a 1-week lag period for diarrhea measurements in children aged 6 months—5.5 years in the control arms in the original trial. Shaded bands indicate simultaneous 95% confidence intervals accounting for clustering. Prevalence estimates were predicted under conditions which held all adjustment covariates at fixed representative values (see Methods for details).
(PDF)

**S4 Fig. Predicted enteropathogen prevalence by temperature minimum.** All panels present adjusted models for children approximately 14 months of age in the control, combined water + sanitation + handwashing (WASH), nutrition, and combined nutrition + WASH arms of the original trial. Shaded bands indicate simultaneous 95% confidence intervals accounting for clustering. Prevalence estimates were predicted under conditions which held all adjustment covariates at fixed representative values (see Methods for details).
(PDF)

**S5 Fig. Predicted enteropathogen prevalence by temperature maximum.** All panels present adjusted models for children approximately 14 months of age in the control, combined water + sanitation + handwashing (WASH), nutrition, and combined nutrition + WASH arms of the original trial. Shaded bands indicate simultaneous 95% confidence intervals accounting for clustering. Prevalence estimates were predicted under conditions which held all adjustment covariates at fixed representative values (see Methods for details).
(PDF)

**S6 Fig. Diarrhea and enteropathogen prevalence by above- vs. below-cutoff for total weekly precipitation at 75th and 90th percentile cutoffs.** All panels present adjusted models including an indicator for above 75th (58mm) or 90th (105mm) percentile average weekly precipitation as the independent variable; unadjusted models produced similar results. Error bars present 95% confidence intervals adjusted for clustering. The x-axis is on the log scale. Panel A) includes diarrhea measurements in children aged 6 months—5.5 years in the control arms in the original trial. Panels B-D) include measurements in children approximately 14 months of age in the control, combined water + sanitation + handwashing (WASH), nutrition, and combined nutrition + WASH arms of the original trial. Closed circles in panels B-D indicate the expected most important lag based on enteropathogen-specific incubation times (Appendix C in S1 Text).
(PDF)

**S7 Fig. Diarrhea and enteropathogen prevalence by heavy rainfall.** All panels present adjusted models including an indicator variable for heavy rainfall (total weekly precipitation > 80th (17mm) or 90th (29mm) percentile during the study period); unadjusted models produced similar results. Error bars present 95% confidence intervals adjusted for clustering. The x-axis is on the log scale. Panel A) includes diarrhea measurements in children aged 6 months—5.5 years in the control arms in the original trial. Panels B-D) include measurements in children approximately 14 months of age in the control, combined water + sanitation + handwashing (WASH), nutrition, and combined nutrition + WASH arms of the original trial. Closed circles in panels B-D indicate the expected most important lag based on enteropathogen-specific incubation times (Appendix C in S1 Text).
(PDF)

**S8 Fig. Prevalence ratios for diarrhea and enteropathogen carriage and distance from study households to surface water.** All panels present adjusted models; unadjusted models

produced similar results. The independent variable was a categorical variable for tertiles of distance from each household to the nearest surface water (<165m; 165m to <316m, [3]316m). Error bars present 95% confidence intervals adjusted for clustering. The x-axis is on the log scale. Panel A) includes diarrhea measurements in children aged 6 months—5.5 years in the control arms in the original trial. Panels B-D) include measurements in children approximately 14 months of age in the control, combined water + sanitation + handwashing (WASH), nutrition, and combined nutrition + WASH arms of the original trial.
(PDF)

**S9 Fig. Prevalence ratios for diarrhea and enteropathogen carriage associated with the proportion of land around study households that contained surface water.** All panels present adjusted models; unadjusted models produced similar results. The independent variable was an indicator for whether the proportion of pixels with surface water within 250m, 500m, 750m of each household was above or below the median. Error bars present 95% confidence intervals adjusted for clustering. The x-axis is on the log scale. Panel A) includes diarrhea measurements in children aged 6 months—5.5 years in the control arms in the original trial. Panels B-D) include measurements in children approximately 14 months of age in the control, combined water + sanitation + handwashing (WASH), nutrition, and combined nutrition + WASH arms of the original trial.
(PDF)

**S10 Fig. Predicted diarrhea and enteropathogen prevalence by vapor pressure deficit.** All panels present adjusted models; shaded bands indicate simultaneous 95% confidence intervals accounting for clustering. Pathogen models included measurements in children approximately 14 months of age in the control, combined water + sanitation + handwashing (WASH), nutrition, and combined nutrition + WASH arms of the original trial. Diarrhea model includes measurements in children aged 6 months—5.5 years in the control arms in the original trial. Prevalence estimates were predicted under conditions which held all adjustment covariates at fixed representative values (see Methods for details).
(PDF)

**S11 Fig. Interaction between age category and hydrometeorological risk factors on childhood diarrhea prevalence.** Data are from adjusted models and include measurements in children aged 6 months—5.5 years in the control arms in the original trial. Panels A) and B) Error bars present 95% confidence intervals adjusted for clustering and the x-axis is on the log scale. Panel A) shows an indicator variable for heavy rainfall (total weekly precipitation $> 80^{th}$ (17mm) or $90^{th}$ (29mm) percentile during the study period or for above median (13mm), $75^{th}$ (58mm) or $90^{th}$ (105mm) percentile average weekly precipitation as the independent variable. Panel B) The independent variable was an indicator for whether the proportion of pixels with surface water within 250m, 500m, 750m of each household was above or below the median or a categorical variable for tertiles of distance from each household to the nearest surface water (<165m; 165m to <316m, $\geq$316m). Panel C) Shaded bands indicate simultaneous 95% confidence intervals accounting for clustering. Prevalence estimates were predicted under conditions which held all adjustment covariates at fixed representative values (see Methods for details). No significant results were observed for other temperature variables or vapor pressure deficit.
(PDF)

**S12 Fig. Negative control analysis–bruising and temperature.** All panels present adjusted models for weekly minimum, mean, or maximum temperature with the indicated lag period; shaded bands indicate simultaneous 95% confidence intervals accounting for clustering.

Includes measurements in children aged 6 months—5.5 years in the control arms in the original trial. Prevalence estimates were predicted under conditions which held all adjustment covariates at fixed representative values (see Methods for details).
(PDF)

**S1 Table. Bacterial enteropathogen prevalence estimates at weekly average temperatures of 19˚C and 30˚C.**
(PDF)

**S2 Table. Akaike Information Criterion (AIC) from adjusted models including temperature and precipitation variables and with or without interaction terms for diarrhea in the previous 7 days or detection of an enteropathogen.** Blue boxes indicate where models without the interaction term had lower AIC while orange boxes highlight models where including the interaction term results in lower AIC. Bolded values emphasize the model with the lowest AIC for the given temperature variable.
(XLSX)

**S3 Table. Adjusted prevalence ratios for caregiver-reported child bruising in the prior 7 days.**
(PDF)

## Acknowledgments

We thank the study participants for their time and sample donation, and hospitality during household sample collection. We also acknowledge the Stanford Research Computing Center for computational resources at the Sherlock high-performance cluster and the International Centre for Diarrhoeal Disease Research, Bangladesh for institutional and administrative support for the field data collection.

## Author Contributions

**Conceptualization:** Jessica A. Grembi, Jade Benjamin-Chung.

**Data curation:** Jessica A. Grembi, Anna T. Nguyen, Stephanie Djajadi, Md Abdul Karim, Md Ohedul Islam, Rana Miah, Syeda L. Famida, Md Saheen Hossen, Palash Mutsuddi, Shahjahan Ali, Md Ziaur Rahman, Zahir Hussain, Abul K. Shoab, Jade Benjamin-Chung.

**Formal analysis:** Jessica A. Grembi, Anna T. Nguyen, Jade Benjamin-Chung.

**Funding acquisition:** Jade Benjamin-Chung.

**Investigation:** Jessica A. Grembi, Anna T. Nguyen, Md Abdul Karim, Md Ohedul Islam, Rana Miah, Syeda L. Famida, Md Saheen Hossen, Palash Mutsuddi, Shahjahan Ali, Md Ziaur Rahman, Zahir Hussain, Abul K. Shoab.

**Methodology:** Jessica A. Grembi, Anna T. Nguyen, Yoshika Crider, Andrew Mertens, Benjamin F. Arnold, Adam Bennett, Jade Benjamin-Chung.

**Project administration:** Jessica A. Grembi, Ayse Ercumen, Audrie Lin, Rashidul Haque, Mahbubur Rahman, Leanne Unicomb, Stephen P. Luby, Benjamin F. Arnold, Jade Benjamin-Chung.

**Resources:** Jessica A. Grembi, Ayse Ercumen, Audrie Lin, Yoshika Crider, Andrew Mertens, Md Abdul Karim, Md Ohedul Islam, Rana Miah, Syeda L. Famida, Md Saheen Hossen, Palash Mutsuddi, Shahjahan Ali, Md Ziaur Rahman, Zahir Hussain, Abul K. Shoab,

Rashidul Haque, Mahbubur Rahman, Leanne Unicomb, Stephen P. Luby, Jade Benjamin-Chung.

**Software:** Jessica A. Grembi, Anna T. Nguyen, Marie Riviere, Gabriella Barratt Heitmann, Arusha Patil, Tejas S. Athni, Benjamin F. Arnold, Adam Bennett.

**Supervision:** Jessica A. Grembi, Ayse Ercumen, Audrie Lin, Shahjahan Ali, Md Ziaur Rahman, Zahir Hussain, Abul K. Shoab, Rashidul Haque, Mahbubur Rahman, Leanne Unicomb, Stephen P. Luby, Jade Benjamin-Chung.

**Visualization:** Jessica A. Grembi, Marie Riviere, Gabriella Barratt Heitmann, Arusha Patil, Tejas S. Athni, Jade Benjamin-Chung.

**Writing – original draft:** Jessica A. Grembi, Jade Benjamin-Chung.

**Writing – review & editing:** Anna T. Nguyen, Marie Riviere, Gabriella Barratt Heitmann, Arusha Patil, Tejas S. Athni, Stephanie Djajadi, Ayse Ercumen, Audrie Lin, Yoshika Crider, Andrew Mertens, Md Abdul Karim, Md Ohedul Islam, Rana Miah, Syeda L. Famida, Md Saheen Hossen, Palash Mutsuddi, Shahjahan Ali, Md Ziaur Rahman, Zahir Hussain, Abul K. Shoab, Rashidul Haque, Mahbubur Rahman, Leanne Unicomb, Stephen P. Luby, Benjamin F. Arnold, Adam Bennett.

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
