## [Decision Letter · Decision Letter 0]

30 Oct 2023

Dear Dr. Grembi,

Thank you very much for submitting your manuscript "Influence of weather and environmental risk factors on child diarrhea and enteropathogens in rural Bangladesh" for consideration at PLOS Neglected Tropical Diseases. As with all papers reviewed by the journal, your manuscript was reviewed by members of the editorial board and by several independent reviewers. In light of the reviews (below this email), we would like to invite the resubmission of a significantly-revised version that takes into account the reviewers' comments. 

We cannot make any decision about publication until we have seen the revised manuscript and your response to the reviewers' comments. Your revised manuscript is also likely to be sent to reviewers for further evaluation.

Sincerely,

Marilia Sá Carvalho

Academic Editor

Stuart Blacksell

Section Editor

Reviewer's Responses to Questions

**Key Review Criteria Required for Acceptance?**

**Methods**

-Are the objectives of the study clearly articulated with a clear testable hypothesis stated?

-Is the study design appropriate to address the stated objectives?

-Is the population clearly described and appropriate for the hypothesis being tested?

-Is the sample size sufficient to ensure adequate power to address the hypothesis being tested?

-Were correct statistical analysis used to support conclusions?

-Are there concerns about ethical or regulatory requirements being met?

Reviewer #1: -The study objectives are clear and the methods are appropriate and well-presented. 

-Line 202-203: I am not sure I agree with your conclusion that pair-wise correlations between weather exposures were not strong and thus not necessary to adjust individual exposure models for other weather exposures. This makes sense for surface water distance and perhaps also for vapor pressure deficit, but I don’t think it makes sense for temperature and precipitation. I would consider correlation coefficients around 0.6-0.7 (as you show in Fig S1) to be moderate or strong, and this makes sense - temperature and precipitation are typically correlated. I would consider looking at the effects of temperature and precipitation in combined models, or address this further and provide a justification for not doing so. 

- If you do include temp and precip in combined models, you might also consider testing whether the inclusion of an interaction term between temp and precip improves the model.

-How did you select these exposure variables? E.g. duration of lag periods, heavy rainfall as >80th percentile, weekly precip over the median. Recommend including a justification. Are the results consistent if these specifications change? It might be valuable to include sensitivity analyses in the supplemental. I see you analyzed 1, 2, and 3 week lags. What about the choice of the 80th percentile and the median precipitation?

-Line 323: How were 250m, 500m, 750m distance thresholds selected?

-Line 232: Could you provide a citation or justification for this method of determining the rainy season?

-The authors have clearly made an effort to make their research process transparent (publicly available deidentified data, code, and pre-specified analysis plan). This is commendable. For the code - you state that code is available at https://github.com/jgrembi/WBB-weather-diarrhea-pathogens, but I get an error message when I try to access this. I am assuming you intend to make it available post publication?

Reviewer #2: The manuscript reports an analysis of secondary data that was originally collected to test the efficacy of WASH and nutrition interventions on childhood diarrhea and growth. The objective is stated clearly at the end of the introduction, though it is not restated as a testable hypothesis, perhaps because of the complexity of the analysis and the multiple pairwise exposure/outcome associations tested. If stating a hypothesis is a requirement, perhaps it could be something like “there are pathogen-specific differences in the direction, magnitude, and significance of associations between environmental exposures and enteropathy outcomes”.

Broadly speaking, the study design is appropriate and follows approaches that have precedents in prior literature. The authors match publicly available hydrometeorological exposure data spatiotemporally with epidemiological outcome data collected by the trial. Unadjusted and adjusted associations are characterized by fitting GAM models, while giving appropriate consideration to lagged effects, spatial autocorrelation, and potential confounding. Notably, they link to their pre-analysis plan and document deviations from it in an appendix, a level of transparency that is rare.

I think that the structure of the dataset and its implication for the model specification could be better explained. It seems to me that the data is hierarchical, with stool samples/diarrhea episodes from multiple rounds nested within subjects (children), subjects nested within compounds, and compounds within villages (clusters). Random intercepts were used in the models to account for this clustering but at different levels depending on the outcome (compound-level for diarrhea, village-level for enteropathogens). I’m sure there were good reasons for treating the outcomes differently, but the justification is not clear as currently written. Were subject-level random effects specified in all models? 

Were the stool samples in which enteropathogens were diagnosed all collected during diarrheal episodes, or were there samples from asymptomatic children?

The use of child bruising as a negative control outcome is appropriate since it can’t plausibly be linked with the exposures or outcomes, however I disagree that the reason for this is “to detect potential misclassification of diarrhea”. The purpose of a negative control outcome is to infer causality with greater confidence (by demonstrating the specificity of the association) and to rule out measured and unmeasured confounding as an explanation for any statistically significant main associations observed (i.e., if an association between an exposure and an NCO is significant, then the main association may be spurious).

I think that the authors’ choice of climate data sources requires some justification. FLDAS was generated for Central Asia, specifically Afghanistan, and not for Bangladesh. The resolution of the NOAA precipitation dataset used is extremely coarse at 55km2. It’s difficult to tell from figure 1 (perhaps include a scalebar) but probably all the clusters are covered by a handful of grid squares, while precipitation varies on a very small scale. The CHIRPS precipitation dataset (https://doi.org/10.1038/sdata.2015.66) has much higher resolution, but also consider the ERA5 dataset (https://doi.org/10.1002/qj.3803), which has been used in epidemiological studies of infectious syndromes (https://doi.org/10.1038/s41467-021-25914-8) and includes many of the variables relevant to this analysis (https://doi.org/10.1038/s41597-023-02276-y), with the advantage that they are from a single source and mutually consistent.

The explanation for the aggregation and lagging of the meteorological exposures is a little unclear. Was the resolution of both the enteropathogen and diarrhea variables daily? At points it sounds like for diarrhea it was weekly. In any case, I think you should state in the main text that two of the main exposures had daily resolution (temperature and precipitation) and the other had monthly resolution (surface water and VPD). Those with daily resolution were aggregated in different ways over a 7-day window (as well as a 30-day window for temperature), and lagged by 7, 14, and 21 days. So, if t0 is the day of outcome ascertainment, then one of the exposures was the average temperature from t-7 to t-14 (see notation used in 10.1029/2021GH000452). The monthly exposures were linked to the outcomes by calendar month and year of outcome ascertainment (without lagging). If I’ve understood this correctly, then I think it might be possible to tighten this explanation. This is especially important for the surface water data as I expect it will not be familiar to readers.

Some additional details of the modeling approach appear to be missing. Please state the link function and family of the model used. From what I can infer, the outcomes were binary (presence or absence of a diarrhea episode/enteropathogen), so would that have been a log link and binomial family? Were prevalence and prevalence ratios approximated from transformations of odds ratios or risk ratios?

If I’ve understood appendix 5 correctly, there were a base set of non-environmental exposures which were included in all models to adjust for their potential confounding effects. This is appropriate. Then for each environmental exposure, a single specific other confounder was considered (but not in the case of precipitation and temperature). Since monthly precipitation was considered as a confounder for two exposures, why not fit a fully adjusted model, testing the associations of all 5 risk factors in table S5 in the presence of each other? (The answer to this question may be that you would need a way of choosing which lag and aggregation window length to use for each exposure/outcome pair and this would overcomplicate the analysis).

Has the distribution of animal ownership by compound been reported anywhere? The use of this variable seems to me to be a bit of a buried lead, and I think a lot of readers would be interested to see the main effect of that variable on the outcomes, or its interaction with the environmental exposures. I do understand, however, that this is beyond the scope of this analysis.

Lines 194 – 195: I’m a little unsure about the justification for dealing with study arm differently for diarrhea and pathogens. Are the results different if you include study arm as a covariate for the diarrhea models, or exclude the intervention arms for pathogens? It seems like a common approach to both outcomes would be fine and would make results more comparable.

Lines 199 – 200: I suggest “(continuous exposures modeled with cubic splines, categorical exposures modeled as factors relative to a reference category)”.

Reviewer #3: The methodology lacks clarity and structure. 

First I suggest describing at the beginning of the methodology the study design used in order to correctly interpret the association measures evaluated in the study.

I suggest adding information on the country, what percentage of the population is represented in the sample of this research, as well as the representativeness of each of the provinces sampled.

I suggest incorporating more information about the climate of the country and the provinces of the study. For example, the monsoon from June to September that often causing flooding, Hurricane season can occur from May to June and October to November, month with high temperatures and humidity and the period of time with major storms.

Why were trials conducted on pregnant women?

I consider that Figure 1 should show the provinces sampled with the continental location of Bangladesh and the left panel I suggest to put it in a separate figure later when the associations with climate are presented. 

Why was the subsample taken in that year (2014)? 

Although the sampling information is detailed in supplementary material, I suggest clarifying some issues in the body of the manuscript:

1º At what scale was the climate data worked on?

2ºHow many weather stations do you work with?

3º At what time of the year were the samples taken, given the intra-annual climatic variability?

4º In the case of the environmental variable: only natural or artificial water bodies were recorded (for example, ponds in parks, drains or containment pools).

Proximity to sources of garbage accumulation could have been another interesting environmental variable to take into account and open dumps can be recorded through spectral signatures as well.

Line 212: “We assessed effect modification for the diarrhea outcome by child age”. Why?

**Results**

-Does the analysis presented match the analysis plan?

-Are the results clearly and completely presented?

-Are the figures (Tables, Images) of sufficient quality for clarity?

Reviewer #1: -The results are clearly presented and the figures are effective.

-Lines 192-195: In your pre-specified analysis plan, you state that “We will pool across all intervention arms from the original study to maximize statistical power and will include study arm as a covariate to adjust for any effects of study interventions on each outcome, except for the diarrhea outcome where we will conduct subgroup analyses separately within the control arm or intervention arms (all grouped together) due to the significant influence of the interventions on diarrhea in a seasonally-dependent manner.” However, in the manuscript, it seems like you exclude diarrhea data for the intervention arms. Please describe if and why this was changed.

-Line 337: Where are these results?

-S1 Appendix 2: I don’t see these additional results at the link included.

Reviewer #2: Line 221: Was the difference in diarrhea incidence in the rainy vs the dry season statistically significant?

Line 258: When you go from describing the distribution of the variables to reporting predictions of the model results, I would start a new paragraph and start it with “According to model results…” or something like that. If I’ve understood correctly, the prevalences reported in table S1 and lines 258 – 264 are predictions from GAMs at two values for temperature at either extreme of the range (and statistically significant results are in bold typeface in table S1). This is not clearly stated, however.

I assume that rotavirus was excluded as a pathogen-specific outcome because it was detected in fewer than 10% of samples (same for astrovirus). Since rotavirus is a highly climate sensitive pathogen and readers will be looking for that result, I think that this should be stated somewhere (e.g., in the limitations paragraph).

In line 285, an adjusted prevalence ratio (aPR) is reported, but after that, the abbreviation “PR” is used suggesting that the rest of the results are unadjusted. Similarly, in the coefficient plots, unadjusted PRs are plotted and the figure legends state that adjusted results are similar. I think it makes more sense to consistently report the adjusted results.

What possible causal explanation could there be for why aEPEC has a negative association with rainfall, but tEPEC has a positive one? I strongly suspect that this is the result of confounding. aEPEC, tEPEC a

---

## [Editor Report · Decision Letter 1]

18 Apr 2024

Dear Dr. Grembi,

We are pleased to inform you that your manuscript 'Influence of hydrometeorological risk factors on child diarrhea and enteropathogens in rural Bangladesh' has been provisionally accepted for publication in PLOS Neglected Tropical Diseases.

Best regards,

Marilia Sá Carvalho

Academic Editor

Stuart Blacksell

Section Editor

---

## [Editor Report · Acceptance letter]

6 May 2024

Dear Dr. Grembi,

We are delighted to inform you that your manuscript, "Influence of hydrometeorological risk factors on child diarrhea and enteropathogens in rural Bangladesh," has been formally accepted for publication in PLOS Neglected Tropical Diseases.

Best regards,

Shaden Kamhawi

co-Editor-in-Chief

Paul Brindley

co-Editor-in-Chief
